# Retention of Phthalates in Wine Using Nanomaterials as Chemically Modified Clays with H_20_, H_30_, H_40_ Boltron Dendrimers

**DOI:** 10.3390/nano13162301

**Published:** 2023-08-10

**Authors:** Andreea Hortolomeu, Diana-Carmen Mirila, Ana-Maria Georgescu, Ana-Maria Rosu, Yuri Scutaru, Florin-Marian Nedeff, Rodica Sturza, Ileana Denisa Nistor

**Affiliations:** 1Department of Chemical and Food Engineering, Faculty of Engineering, “Vasile Alecsandri” University of Bacau, 157 Calea Marasesti, 600115 Bacau, Romania; hortolomeuandreea@gmail.com (A.H.); miriladiana@ub.ro (D.-C.M.); ana.georgescu@ub.ro (A.-M.G.); ana.rosu@ub.ro (A.-M.R.); 2Department of Oenology and Chemistry, Faculty of Food Technology, Technical University of Moldova, 9/9 Studentilor Street, MD-2045 Chisinau, Moldova; iurie.scutaru@enl.utm.md (Y.S.); rodica.sturza@chim.utm.md (R.S.); 3Department of Environmental Engineering and Mechanical Engineering, Faculty of Engineering, “Vasile Alecsandri” University of Bacau, 157 Calea Marasesti, 600115 Bacau, Romania; florin_nedeff@ub.ro

**Keywords:** oenology, bentonite, dendrimers, phthalates, polyphenols, white wine

## Abstract

The presence of phthalic acid esters in wines presents a major risk to human health due to their very toxic metabolism. In this paper, aluminosilicate materials were used, with the aim of retaining various pollutants and unwanted compounds in wine. The pollutants tested were di-butyl and di-ethyl hexyl phthalates. They were tested and detected using the gas chromatography–mass spectrometry (CG-MS) analytical technique. Nanomaterials were prepared using sodium bentonite, and were chemically modified via impregnation using three types of Boltron dendrimers of second, third and fourth generations (NBtH_20_, NBtH_30_ and NBtH_40_). The synthesized nanomaterials were characterized using the Brunauer–Emmett–Teller (BET) method, Fourier-transform infrared spectroscopy (FTIR) and X-ray diffraction (XRD) analysis. In this paper, two aspects were addressed: the first related to the retention of phthalate-type pollutants (phthalic acid esters—PAEs) and the second related to the protein and polyphenol levels in the white wine of the Aligoté grape variety. The results obtained in this study have a major impact on PAEs in wine, especially after treatment with NBtH_30_ and NBtH_40_ (volumes of 250–500 μL/10 mL wine), with the retention of the pollutants being up to 85%.

## 1. Introduction

Wine is an alcoholic beverage made up of 80% water, 12–15% ethyl alcohol, and a minor amount of constituents, namely 3% (acetaldehyde, glycerol, tartric acid, lactic acid and malic acid) [1]. Other compounds that can be present in wine are as follows: organic acids, sugars, polyphenolic compounds, nitrogenous compounds, enzymes, vitamins, lipids, volatile compounds, etc. Among the minor compounds in white wine, the most important are organic acids and phenolic compounds [2]. They significantly affect the quality of the wine from an organoleptic point of view [2,3]. Some wine constituents are able to react with large amounts of oxygen, but polyphenols are the most prone to oxidation [4]. The concentration of flavonoids in wine is strongly affected by the following stages of winemaking: pressing and maceration. They can affect the degree of extraction from the skin of the grape berries, but especially from the seeds, due to the high content of proanthocyanidins. The contents of the previously mentioned compounds can be found in white wine up to the level of 100 mg·L^−1^ [4] White wines have a wide variety of bio-polymeric compounds in their composition. Obtaining stable wines is achieved by reducing, as much as possible, the bio-polymeric compounds, because they are responsible for the instability of the final product (wine). To solve this impediment, winemakers resort to different treatment processes using materials capable of absorbing the unwanted protein compounds from the wine.

Phthalates (PAEs) are among the desirable compounds found in the materials used in the wine industry, and have a disruptive effect [5]. PAEs are added to the mass of plastics due to their flexibility and durability [6]. The molecular weight of these compounds is small and they are formed via the reaction of phthalic anhydride with linear or branched alcohols [7]. Phthalate migration rate studies of polymer coatings in food environments have demonstrated that PVC and rubber retain their ability to release a phthalate even after a long period of use.

The migration rate of phthalates in plastics depends on the chemical composition of the extraction medium. Environments with high polarities are the most contaminated, especially in cases in which the thermal factor intervenes. Wines are among the most frequently affected [8,9]. It has been found that, of the variety of PAEs, the most common, both in wine and in other food products, are those of the di-ethyl hexyl phthalate type (DEHP, shown in Figure 1a) [10,11,12,13,14,15] and di-butyl phthalate type (DBP, shown in Figure 1b) [10,12,13,14,16,17,18].

They significantly affect the health of the consumer. In their 99% purified form, PAEs are in the form of viscous, transparent, low-volatile, colorless, odorless, hydrophobic organic liquids under normal conditions, are insoluble in water, and have a high affinity for alcoholic solutions [5]. The daily intake of PAEs tolerated and established by the European Food Safety Authority (EFSA) is 50 μg·kg^−1^bw for DEHP [19] and 10 μg·kg^−1^bw for DBP [20,21,22]. The most common method for the identification and quantification of low concentrations of phthalates in alcoholic beverages is gas chromatography coupled to mass spectrometry (GC–MS) [5,12,14,18,21,23].

**Figure 1 nanomaterials-13-02301-f001:**
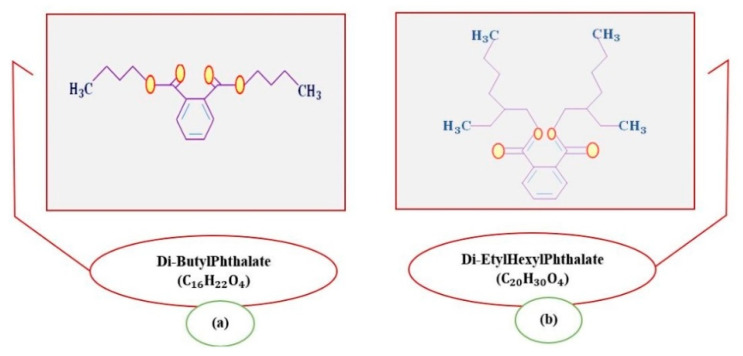
Representation of the chemical structure of Di-Butyl Phthalate (DBP) and (**a**) Di- Ethyl-Hexyl Phthalate (DEHP) (**b**) adapted from [22].

Wine forms tartaric salts with potassium or calcium ions, and this factor leads to the formation of a wine defect from an aesthetic point of view [23]. A solution to prevent such wine breakdowns would be the use of clay-absorbent materials enhanced with different organic compounds such as dendritic polyols. Among the materials with a stabilizing role, aluminosilicates are the most common [24]. Sodium bentonite is frequently used at the industrial level [25,26,27,28]. In the final product (wine), in addition to proteins, other unwanted compounds are present, including tartaric acid, malic acid, etc. Via the contact of white wine with the different materials used during the fermentation and maturation processes (corks, gaskets from fermentation tanks, rubber hoses), there is a risk of changing the composition of the final product [23].

Boltron-type dendrimers are dendritic polymers characterized by an intensely branched architecture with a large number of functional groups. Dendrimer structures are differentiated by their core, which is represented by polyalcohols or hydroxy acids [29,30]. The basic compounds obtained from these complex structures are hydroxyl-functional dendritic polyesters [25,26]. These dendrimers are formed via the polymerization of the particular core and of 2,2-dimethylol propionic acid. These polymers are known as bis-MPA, aliphatic compounds with tertiary ester bonds [31]. Dendrimers present certain specific characteristics, such as high thermal and chemical properties, solubility, complexation capacity, and compatibility. All these characteristics of dendrimers indicate the possibility of their use in the wine industry. Dendrimers have been used in winemaking as encapsulating agents to capture and remove tartaric acid (TA) from white and red wines [28,29,30]. The formation of potassium or calcium tartrate salts leads to the obtainment of visible crystals in the wine, this aesthetic defect not being desirable in the final product. By binding tartaric acid to the interior of the dendritic polymer, a dendrimer–tartaric acid-type complex is formed; this can be removed from the wine via ultrafiltration or reverse dialysis [32]. More recently, dendrimers have been used as the stabilizers of anthocyanins in young red wines (e.g., malvidin-3-glucoside, cyanidin-3-glucoside) [33]. Another utility of polymers is in the identification of metal ions in red wines, such as Pb(II), Co(II), Cu(II), Fe(III), and Zn(II) [34]. This alternative has been used for the capture of tartaric acid from white wine using different organic solutions, with different dendrimers used as encapsulating agents [32,35,36,37,38].

The aim of this paper is to highlight the effect of chemically modified natural materials on the adsorption of phthalates from white wines. The raw material used for intercalation with Boltron dendrimers was sodium bentonite due to its availability in nature, its low operating cost, and its good ability to absorb the colloidal dispersions present in the wine [39,40,41,42,43]. The contact of the nanomaterials and their use in the retention of phthalates represent important goals in the application of advanced materials in the food industry. The novelty of this article is its use of a cationic clay intercalated with Boltron dendrimers of different generations (second, third and fourth generations) (Figure 2), employed in the food industry as advanced materials that retain the phthalates present in alcoholic beverages such as white wine.

By treating white wine with hybrid organic–inorganic materials (bentonite–dendrimers), the effect of these nanomaterials on proteins and polyphenolic compounds in the wine was studied. This was performed by applying the thermal stability test, spectrophotometric analysis and making turbidity measurements. Another aspect monitored was the capture degree of DBP and DEHP phthalates from Aligoté white wine, identified with the help of gas chromatography coupled to mass spectrometry (GC–MS).

## 2. Materials and Methods

### 2.1. Materials

A white wine from the European Aligoté grape variety was selected for research. It was obtained in 2019, following a general technological process for white wines. The raw material was cultivated and processed at the Microvinification of the Department of Oenology, Technical University of Moldova, Republic of Moldova. The wine sample was filtered with a 0.45 μm microfilter and determined spectrophotometrically using the Analytik Jena Specord 250 Plus UV–Vis device. As polluting materials, two standard solutions of high-purity PAEs were chosen for this study, namely di-butyl phthalate (DBP, 99.8%) and dihexyl phthalate (DEHP 99.7%), both purchased from Sigma-Aldrich (Darmstadt, Germany).

The raw material used to obtain the adsorbent materials (NBtH_20_, NBtH_30_, and NBtH_40_) was sodium bentonite Fluka (NBt) procured from Sigma-Aldrich, Germany. This was intercalated with Boltron dendrimers (second, third and fourth generation) purchased from Perstorpt Polyol (Toledo, OH, USA). Acetone and 70% ethyl alcohol were purchased from Sigma-Aldrich.

### 2.2. Modification of Clay-Based Material

Second-, third- and fourth-generation dendrimer solutions were obtained by dissolving 347 mg of the dendrimer of each generation in 10 mL of water/ethanol solution (in a 1:1 mass ratio) at room temperature and under constant stirring for 5 h. Then, 10 mL of the prepared dendrimer solution was added drop by drop over 1.5 g of NBt. After the intercalation of bentonite with each organic solution, they were placed in the thermostat at 308 K for 3 days, discontinuously, with resting periods of 7.5 h at 295 K. The nanomaterials obtained based on the clay chemically modified with dendrimers were further named NBtH_20_, NBtH_30_ and NBtH_40_.

### 2.3. Preparation of Phthalic Solutions and Synthetic Solutions of Pollutants

Two synthetic phthalic solutions of di-butyl and di-ethylhexyl were prepared. From each phthalate solution (>99% concentration), 20 μL was taken. The sampled solution was placed in one Eppendorf tube, with each tube containing 1 mL of 70% ethanol solution. In the end, two different phthalic solutions of 2% concentration (DBP 2% and DEHP 2%) were obtained, which were immersed for 13–15 h at 276 K.

For the preparation of 17 wine samples, 10 mL of filtered Aligoté wine was used. The samples were prepared for synthetic contamination at room temperature (292–296 K). Then, 20 μL of 2% DBP solution and 20 μL of 2% DEHP solution were added to each white wine sample. After the addition of phthalates to the wine samples, the entire sample was shaken using a Hettich EBA200 centrifuge at 200 rpm for 5 min. After this time, the samples were left to rest for 15–20 min at room temperature. After resting, the samples were divided into four wine samples. These were put in contact with the sodium bentonite and with the three chemically modified nanomaterials via impregnation with Boltron dendrimers. For each clay adsorbent material, four white wine samples were allocated. The 17th wine sample was considered the control sample.

The first mini-sample was treated with the raw material (NBt), the second with NBtH_20_, the third with NBtH_30_ and the fourth with NBtH_40_. The adsorbent materials used (5% concentration) were added to each wine sample in different volumes: 50, 100, 250 and 500 μL of adsorbent nanomaterial/10 mL wine. After adding the adsorbent materials, the 4 mini-samples of wine and the control sample were centrifuged (200 rpm, for 10–15 min), then immersed for 48 h in the cold (275–276 K).

### 2.4. Determination of the Degree of Protein Stability

After 48 h of cold immersion, the synthesized wine samples were left at room temperature for 35 min. Next, the control sample and the 4 series of wine treated with the four adsorbent nanomaterials (NBt, NBtH_20_, NBtH_30_, NBtH_40_) were heated at 333 K for 60 min, then cooled at 295 K. To test the protein level in the treated wine, 5 mL of wine sample was extracted from each mini-sample. The amount of sample collected was filtered with a 0.45 μm microfilter and analyzed nephelometrically, spectrophotometrically and via gas chromatography coupled to mass spectrometry (CG-MS).

### 2.5. Characterization of the Prepared Nanomaterials

The nanomaterials prepared in this study were characterized via the Brauner–Emmet–Teller (BET) method, using the NOVA 2200e Gas Sorption Analyzer (Quantachrome, Boynton Beach, FL, USA). This method of analysis enables the characterization of porous materials via the determination of the specific surface area, the volume occupied by the pores, the diameter and the pore size distribution.

Fourier-transform infrared spectroscopy (FTIR) analysis was performed using an Agilent Technologies Cary 630 FTIR spectrometer (Santa Clara, CA, USA). This method revealed the presence of bonds between clay and dendrimers with the aim of obtaining a complex material.

Another characterization method used was X-ray diffraction (XRD). This was performed using the MiniFlex 300/600-type device (Rigaku Corporate, Tokyo, Japan) (40 kV, 15 mA, 2 deg/min, X-ray tube-Cu). This method highlights the basal distance of the tested materials, as well as their mineralogical composition.

To determine the phthalate level in the wine samples after different treatments, the Shimadzu GC MS-QP 2010 SE (Shimadzu Corporation, Kyoto, Japan) apparatus was used (RESTEK Rtx-5 MS capillary column (Thermo Fisher Scientific, Waltham, MA, USA), 30 m/0.25 mm/0.25 μm; EPA 506 Phthalate Mix standard solution, 1000 μg·mL^−1^ in isooctane, purchased from Sigma-Aldrich (Darmstadt, Germany), organic solvent for extraction—CHCl_3_—purchased from Chem-Lab, Singapore).

In order to measure the turbidity of the wine samples, the WTW Labor-type Turb 555 turbidimeter (Thermo Fisher Scientific Inc., Berlin, Germany) (range 0–500 NTU) was used.

All calculations were performed using Microsoft Office Excel 2007 (Microsoft, Redmond, WA, USA). Data obtained in this study are presented as mean values ± standard error of the mean, calculated for three series of parallel experiments.

## 3. Results and Discussions

### 3.1. BET Analysis

Table 1 shows the values of the specific surface areas (S_BET_) for natural bentonite and for the materials intercalated with dendritic polyol compounds. Through the chemical modification of natural bentonite, it is observed that the specific surface decreases significantly from 18.74 m^2^·g^−1^ to approximately 8 m^2^·g^−1^ for the chemically modified bentonites.

A possible explanation for these major changes is the extrinsic porosity in the interlamellar zone of the organo-clay samples. These results are in agreement with the literature regarding the intercalation of clays with dendrimers [39,44]. These data may indicate that these compounds (polyol-type dendritic polymers) form distinct architectures during the modification process of natural bentonite with a minimal amount of dendritic material, approximately 1%.

The textural properties determined via BET N_2_ adsorption analysis for the raw material and for the three synthetized nanomaterials (NBtH_20_, NBtH_30_, and NBtH_40_) are presented in Figure 3.

The nitrogen adsorption–desorption isotherms are similar for all fourth adsorbent materials. According to the IUPAC classification, they are of type IV, being specific to mesoporous adsorbents. Hysteresis occurs due to the fact that the adsorbed molecules do not desorb as easily as they were adsorbed. The hysteresis loop is of the H3 type, indicating the presence of aggregates of flat particles or adsorbents containing slit-shaped pores.

### 3.2. DRX Analysis

The incorporation of organic compounds with the three types of dendritic solutions in the sodium bentonite samples induced shape changes and decreased the d_001_ basal distance [45,46]. According to the literature, these changes represent reliable indicators for the rearrangement of the structure of the smectite material and its impregnation with the organic one.

These modifications to the NBt samples using organic fragments (Figure 4) led to a decrease in the d_001_ basal distance, leading lower 2-theta values [44,47]. However, it can be observed that the sharp reflection of 001 XRD decreases; a possible explanation for this is the loss of the micro-porosity level. Impregnation with Boltron organic solutions produced more disordered interlamellar arrangements in the clays [45,46,47].

The intensity of the peaks for montmorillonite (5 degrees zone) decreased from 4.041 for natural bentonite to 3.258 Å for NBtH_40_ after modifying the clay with Boltron dendrimers of different generations. At the same time, the intensity of the quart peaks increased with the addition of dendrimers.

This change is evident starting from the clay sample impregnated with small-generation Boltron-type dendrimers (Figure 4b). One possible reason is the aggregation of the dendrimer in the aluminosilicate material via hydrogen bonds. It can also be said that bentonite favors dendrimer dispersion, regardless of its generation.

Via the chemical modification of clay with dendrimers, other mineral compounds such as albite, analbite, quartz, lithium dihydrogen disilicate were obtained. The lithium dihydrogen disilicate was predominantly found in NBtH_30_.

### 3.3. FTIR Analysis

In the infrared range, different main characteristics were identified with the help of FTIR analysis (Figure 5).

The impregnation of the dendritic solutions in NBt led to the modification of the clays’ structure via an increase in the intensity of the band in the range 3628–3633 cm^−1^; this area is attributed to the presence of vibrations of hydroxyl-type bonds. The bands in the range 3611–3856 cm^−1^ belong to the stretching vibrations of the -OH groups (Table 2) [48].

The other bands may have changed due to the presence of different structural groups of NBt; for example, the bands in the range of 921–1047 cm^−1^ are due to the vibrations of the Si–O–Si groups [48]. It can be highlighted that in the area of the 1009 cm^−1^ band (the Si–O vibration stretching out of the plane), significant changes took place. The aforementioned band maximum shifts to a much higher number with the change in the NBt with each Boltron dendrimer generation type (up to: 1043 cm^−1^ for NBtH_20_, 1047 cm^−1^ for NBtH_30_, 1057 cm^−1^ for NBtH_40_).

A possible explanation for the displacement of the band involves the deformation of the tetrahedral layers of the SiO_4_^−^ fragments of the bentonite, after its impregnation with each dendritic solution of the second, third and fourth generations (Table 2 and Figure 5). This aspect correlates with the results of the DRX diffractograms.

### 3.4. Analysis of Synthesized Wine Samples Using the Nephelometric and UV–Vis Spectroscopic Methods

#### 3.4.1. Analysis of the Synthesized Wine Using the Turbidimetric Method

The synthesized wine samples were analyzed after performing the protein test, at 333 K for 1 h in a water bath while being slowly stirred. After testing, the synthesized wine samples were allowed to cool at room temperature.

The results presented in Figure 6a, representing the case in which the wine was treated with raw bentonite, provide information on the high degree of protein destabilization. The reasons for the destabilization of proteins in wine can be diverse, including temperature, metals, the presence of tannins and pH variation; ethanol can precipitate some non-protein compounds (polysaccharides and tannins) [51]. After treating the wine with the unmodified clay material, the turbidity of the wine exponentially increased after it was treated at a high temperature. A possible explanation for this result is the presence of mannoproteins and phthalic compounds in the Aligoté wine.

In the case of wine treated with clay solutions modified with dendrimers of different generations, a high level of protein adsorption efficiency was observed.

Starting from low concentrations of NBtH_20_, it can be said that wine is stable when treated with this adsorbent nanomaterial with an amount of up to 250 µL for 10 mL of wine. This aspect was also confirmed after performing the heat test of protein. If a volume greater than 250 µL of NBtH_20_ is added to the wine, protein and non-protein compounds are destabilized. In parallel, the same situation was observed for Aligoté wine treated with the NBtH_30_ and NBtH_40_-type adsorbent nanomaterials with an addition of over 100 µL/10 mL of wine sample.

#### 3.4.2. UV–Vis Spectroscopic Analysis

After performing the protein stability test, the synthesized wine samples were filtered through a 0.45 µm cellulose membrane and subjected to UV–Vis spectrophotometric analyses. Certain regions of the absorption spectra (280–360 nm) are of interest, as shown in Figure 7. According to the UV–Vis analysis, the presence of phenolic compounds can be found around the wavelength of 280 nm [52]. Another interval of interest is present in the 300–360 nm region, which is specific to the compounds present in white wines, namely hydroxycinnamic non-flavonoids. In order to assess the extinction coefficients, the curves were plotted. The medium extinction coefficients obtained for the chemically modified bentonite were in the same order, around 0.7 L·mmol^−1^·cm^−1^.

The spectra in the UV–Vis range of white wine (Figure 7) have an absorption maximum in the 265–280 nm range, a characteristic area for phenolic substances (absorption of the aromatic ring in which different radicals can adhere, which can influence the position of the maximum) [52]. Specific majority phenolic compounds (such as hydroxycinnamic compounds and their esters) are found in white wines.

In the absorption spectra, inflection bridges can be observed in the 310–330 nm absorption range, a specific area for cinnamic substances. The decreasing intensity in the 315–330 nm range indicates a lower amount of hydroxycinnamic compounds. After treating the wine with NBtH_30_ and NBtH_40_ at adsorbent material concentrations above 250 µL, a massive decrease in the wavelength range specific to cinnamic substances can be observed. This information is important because a low level of these substances (hydroxycinnamic compounds) decreases the risk of the oxidation process, thus preventing the browning of Aligoté wine and preserving its organoleptic properties.

By analyzing the absorption spectra and comparing them with the spectrum of the untreated wine, a difference can be found between them at various wavelengths, as follows: in the range of 265–275 nm, the wine treated with NBt has its maximum absorption at 260 nm (Figure 7a). Wine treated with NBtH_20_ shifts its absorption maximum to 264 nm (Figure 7b), that treated with NBtH_30_ shifts to 273 nm, and that treated with NBtH_40_ shifts to 267 nm (Figure 7c,d). This information may indicate the formation of oxidized phenolic compounds [53,54].

From the point of view of the displacements of the absorption spectra after the treatment of white wine with dendritic organic adsorbent materials, a difference is observed for NBtH_40_ with an addition of 500 µL in the 308–330 nm region, which is specific to cinnamic compounds.

For the other spectra at the same wavelengths (Figure 7b,c), there are increases in the absorbance when using 500 µL of NBtH_40_ (Figure 7d) at 315–325 nm; a possible explanation for this is the higher level of oxidized polyphenolic compounds. The same is observed for NBtH_30_, as a result of the increase in oxidized polyphenols, followed by NBtH_20_.

The concentrations of phenolic compounds in Aligoté white wine after treatment with clay adsorbents are presented in Table 3. The phenolic substances shown in the table are as follows: total phenolic substances—TPS (expressed in Gallic Acid Equivalent, mg·L^−1^), cinnamic phenolic substances—CPS (expressed in Caffeic Acid Equivalents, mg·L^−1^), and the phenolic flavonoid substances—PFS (expressed in Catechin Equivalents, mg·L^−1^).

The deposition/reduction of a part of the phenolic complex was carried out by using organic adsorbent nanomaterials modified with dendrimers of higher generations (NBtH_30_ and NBtH_40_) in the 250–500 µL concentration range. By analyzing the white wine treated with NBt and comparing it with adsorbents chemically modified with dendrimers, it can be said that the organic adsorbent material works differently on phenolic compounds. This can be seen in Figure 7b–d and in Table 3.

The level of the total polyphenolic index (ITP) decreased by 11.36% after treatment with NBt; after treating Aligoté white wine with adsorbent materials chemically modified with dendrimers, the ITP decreased by 17.05% (with 250–500 µL of NBtH_40_ and NBtH_30_). In the case of TPS, the decrease was up to 28% when using NBtH_40_ and NBtH_20_ materials, compared to NBt when it was 19%. For CPS, a decrease in the hydroxycinnamic groups by 17.32% can be observed when treating the wine with 500 µL of NBtH_40_ (Figure 7d). Compared to NBtH_40_, the treatment of wine with the unmodified material led to an increase in CPS by 6.6%; the same situation can be seen for NBtH_30_, with an increase of 10.5%.

The level of PFS for Aligoté white wine decreased by 57.14% when treated with 100–500 µL of NBtH_20_, and for the NBtH_30_ (500 µL) and NBtH_40_ (100 µL) adsorbents, it decreased by up to 42.8%. This information is valuable because it can indicate the condition of the wine after being treated with the adsorbents used in this paper. This shows the high degree of confidence we may have in adsorbent materials modified with dendrimers for the stabilization of a white wine, by preserving its organoleptic properties.

### 3.5. Determination of Phthalate Concentrations by GC–MS

After the heat treatment and testing of the synthetic wine solutions, the phthalate content of the synthesized wine samples was evaluated using GC–MS analytical techniques. The two phthalates were introduced into each wine sample in equal amounts. In the untreated wine sample, a concentration for each pollutant (DBP and DEHP) of 0.58 mg·L^−1^ was detected (Figure 8a).

Due to the hydrophobic properties of the analyzed phthalates, it was found that the level of DEHP decreased significantly, following the treatment with both NBt and the other modified adsorbent materials (Figure 8). The same can also be observed for DBP, although the level of retention is moderate. Thus, after treating the Aligoté white wine with NBtH_20_ and NBtH_40_ (Figure 8b,d), it can be observed how the phthalic content decreases by over 40% for DBP, and how for DEHP, the retention is over 95% after treatment with the modified adsorbent materials using NBtH_20_ and NBtH_30_ (Figure 8b,c).

After treatment with the adsorbent materials modified with Boltron dendrimers of the fourth generation, an 85% reduction in DEHP can be observed (Figure 8d), and in the case of the NBtH_30_ nanomaterial, a 50% decrease in DBP can be seen (Figure 8c).

A possible explanation for the retention of these pollutants would be the thermal treatment of the wine samples because it could influence the adsorption efficiency of the clay material [53]. Another explanation would be the nature of the cations between the layers of the smectite material because this can influence the adsorption process of PAEs.

The adsorption capacity of the adsorbent materials in the contaminated Aligoté white wine is directly proportional to the molecular weight of the DBP and DEHP. The adsorption of these PAEs depends on their molecular mass and chemical structures [9]. The -OH and Si–O groups in the structures of the four adsorbent materials lead to the formation of hydrogen bonds with PAEs [54,55]. This result leads to the adsorption of phthalic compounds in the clay structure, between the interphyllitic zones of NBt, NBtH_20_, NBtH_30_ and NBtH_40_ through electrostatic attraction and hydrogen bonds.

Another important role for increasing the retention of PAEs is the pH of the solution [54,56,57]. At acidic pH, the molecules of PAEs hydrolyze into phthalic acid, and the carbonyl group (C≡O) reacts with hydrogen ions in the solution of the adsorbent material present in the wine.

The obtained results certify the novelty of this study, including the use of nanomaterials in the adsorption of pollutants from the wine industry. These nanomaterials are known for their application in environmental remediation, thus opening new research prospects regarding the regeneration of materials from industries [46,58,59,60,61,62].

## 4. Conclusions

The intercalation of the sodium bentonite with dendritic polymers led to the structural modification of NBt via the formation of organic clusters in the interlamellar structures. This type of modification of the raw material is confirmed by the results of the analysis carried out: BET, DRX and FTIR. The best yield in the retention of PAEs was 95%, being obtained by using 250–500 µL/10 mL of wine of organo-anorgano adsorbent nanomaterials of the NBtH_30_ and NBtH_40_ type for DEHP. Also, for DBP retention, the yield was 50% after treatment with 500 µL/10 mL of wine of NBtH_20_ and NBtH_30_.After performing the protein test, the turbidity of the wine decreased after treatment with the adsorbent nanomaterials modified using dendrimers with a concentration of 100–250 µL/10 mL of wine. The level of total phenolic compounds decreased by 28% after treating the wine with NBtH_40_, followed by NBtH_20_. Hydroxycinnamic substances decreased by 17% compared to the original, untreated wine. This was achieved with the addition of 500 µL of organic nanomaterials of the highest generation (NBtH_40_). In the case of phenolic flavonoid substances, the decrease was 50% for NBtH_20_ in low concentrations, followed by NBtH_30_ and NBtH_40_ with higher concentrations. It can be concluded that clay minerals modified with Boltron dendrimers are nanomaterials with good structural and morphological properties, and were chosen for their use in the treatment of white wines for phthalate retention. These materials can be used in white wines to reduce polyphenolic and cinnamic compounds.

## Figures and Tables

**Figure 2 nanomaterials-13-02301-f002:**
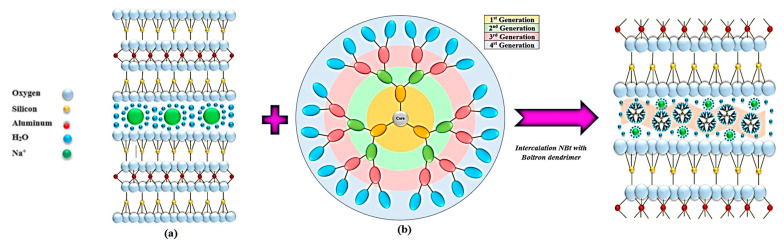
General representation of the impregnation of sodium bentonite NBt (**a**) with Boltron-type dendritic nanomolecules of second, third and fourth generations (**b**).

**Figure 3 nanomaterials-13-02301-f003:**
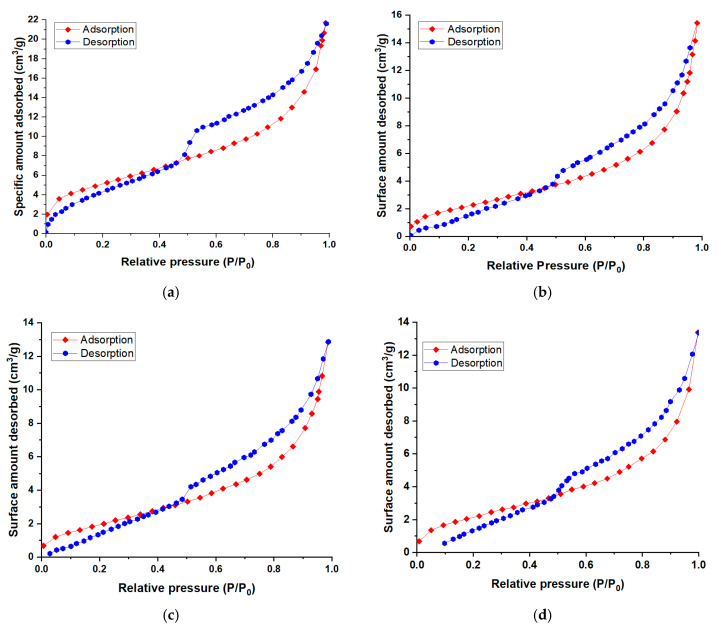
Nitrogen adsorption/desorption isotherms for NBt (**a**), NBtH_20_ (**b**), NBtH_30_ and (**c**) NBtH_40_ (**d**).

**Figure 4 nanomaterials-13-02301-f004:**
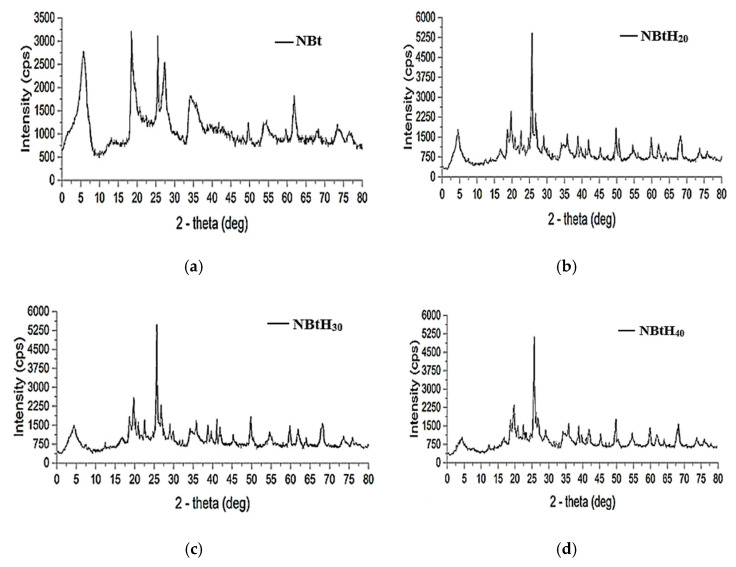
DRX analysis for samples of NBt (**a**) enhanced with Boltron-type dendrimers for second (**b**), third (**c**) and fourth (**d**) generations.

**Figure 5 nanomaterials-13-02301-f005:**
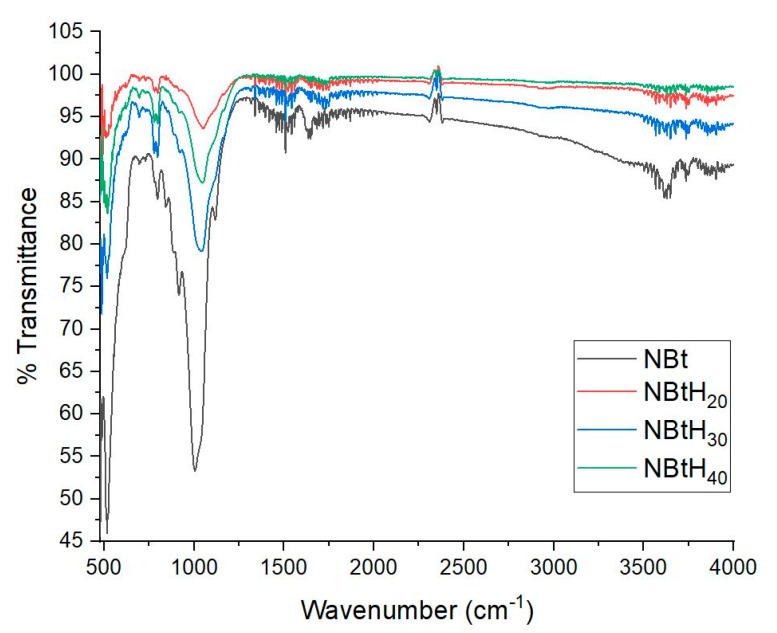
FTIR analysis for NBt (1) samples impregnated with Boltron-type dendrimers for second (2), third (3), fourth (4) generations.

**Figure 6 nanomaterials-13-02301-f006:**
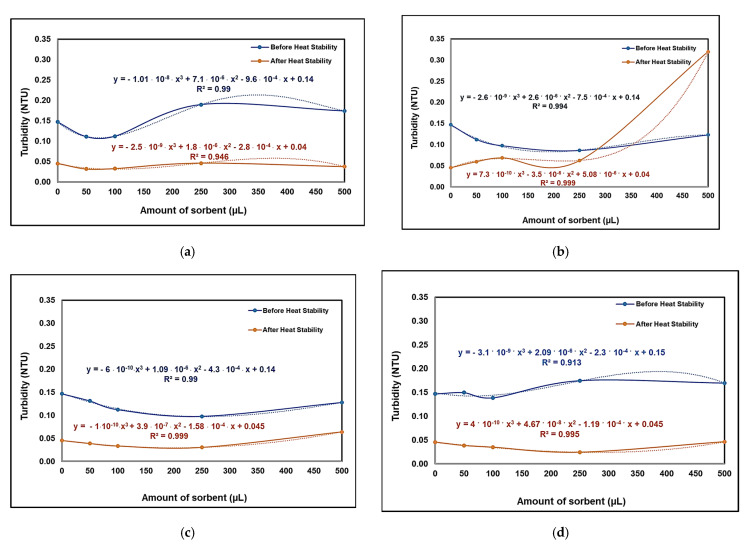
Results of turbidimetric (NTU) tests for Aligoté wine, before and after heating, treated with raw bentonite—NBt (**a**), and bentonite modified with Boltron type-dendrimers of the second (**b**), third (**c**) and fourth (**d**) generations; 10 mL of wine was used in all cases.

**Figure 7 nanomaterials-13-02301-f007:**
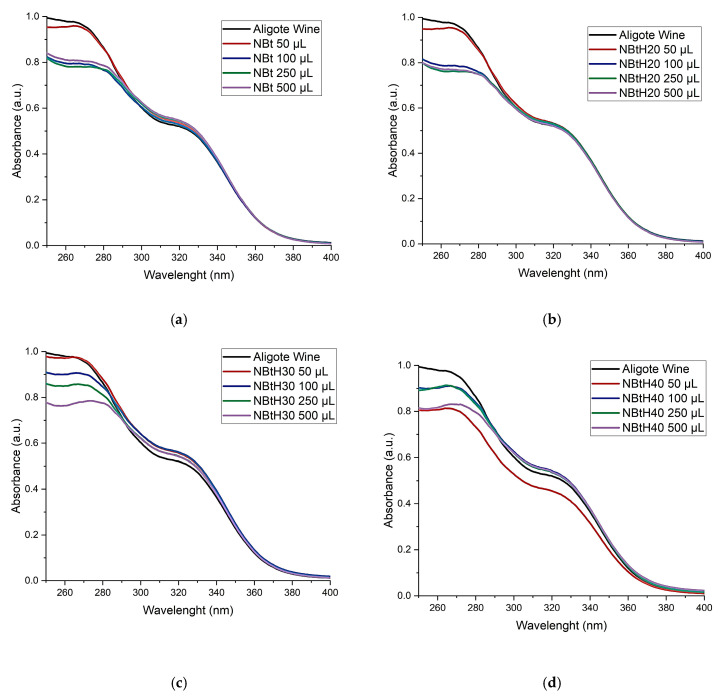
The effect of NBt (**a**), NBtH_20_ (**b**), NBtH_30_ (**c**) and NBtH_40_ (**d**) materials in Aligoté wine: (1) untreated Aligoté wine, (2) Aligoté treated with 50 µL of material/10 mL wine; (3) Aligoté treated with 100 µL of material/10 mL wine; (4) Aligoté treated with 250 µL of material/10 mL wine; (5) Aligoté treated with 500 µL of material/10 mL wine.

**Figure 8 nanomaterials-13-02301-f008:**
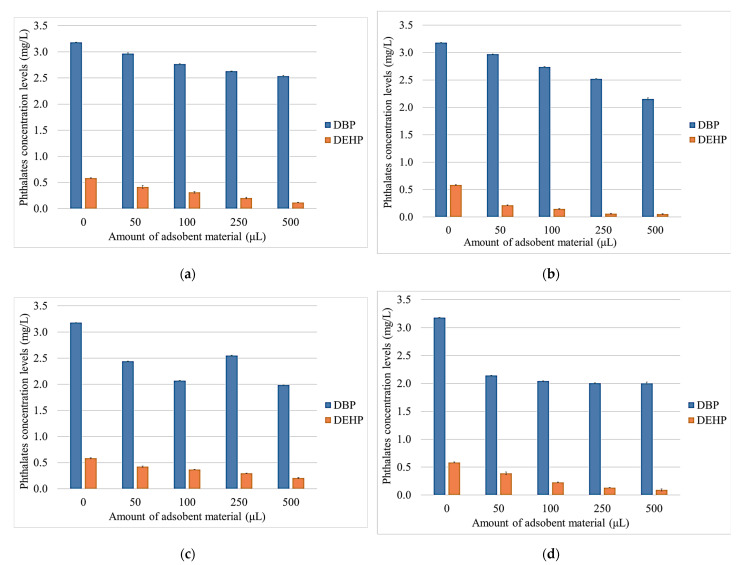
Phthalates concentration levels determined in Aligoté white wine with NBt (**a**), NBtH_20_ (**b**), NBtH_30_ (**c**), NBtH_40_ and (**d**) adsorbent materials.

**Table 1 nanomaterials-13-02301-t001:** Textural analysis of the dendrimer-modified clay samples.

	Sample Name			
	NBt	NBtH_20_	NBtH_30_	NBtH_40_
Specific Surface Area (m^2^·g^−1^)	18.74	8.68	7.84	8.58
Total Pore Volume (cm^3^·g^−1^)	0.034	0.024	0.020	0.021
Average Pore Diameter (nm)	7.18	11.06	10.17	9.67

**Table 2 nanomaterials-13-02301-t002:** FTIR band assignments for chemically modified bentonites [48,49,50].

NBt (cm^−1^)	NBtH_20_ (cm^−1^)	NBtH_30_ (cm^−1^)	NBtH_40_ (cm^−1^)	Assignements
517	557	518	521	Al–O–Si stretching
	648		627	OH bending
694		696	698	Quartz
	712			Free or Amorphous Silica
			785	(Al, Mg)–O–H
		790		Si–O stretching Si–O–Al stretching
801	810			Al–Mg–OH bending
843				
881	895	885	879	Al–Fe–OH bending
921		922	912	Al–Al–OH bending
1005	1059	1028	1047	Si–O–Si, Si–O stretching of silica and quartz Si–O stretching, in plane
1119	1196		1165	Si–O stretching, out of plane
1339		1362	1360	CO_3_ stretching of calcite and dolomite
1638	1655	1656	1647	OH bending, hydration
			3353	OH stretching, hydration
3611	3736 3856	3674 3854	3678 3747	OH stretching
3676				
3736				

**Table 3 nanomaterials-13-02301-t003:** Determination of the concentrations of phenolic compounds in Aligoté wine, untreated and treated with different materials.

Clays Samples	Index of Total Polyphenols—ITP				Total Phenolic Substances—TPS (Galic Acid Eq., mg·L^−1^)			
	50 µL	100 µL	250 µL	500 µL	50 µL	100 µL	250 µL	500 µL
Aligoté	8.65 ± 0.0121				136.77 ± 0.040			
NBt	8.58 ± 0.025	7.63 ± 0.044	7.65 ± 0.069	7.84 ± 0.042	135.64 ± 0.055	107.92 ± 0.045	108.82 ± 0.043	113.87 ± 0.014
NBtH_20_	8.57 ± 0.023	7.61 ± 0.045	7.52 ± 0.026	7.70 ± 0.046	134.38 ± 0.041	105.64 ± 0.057	103.37 ± 0.069	113.28 ± 0.085
NBtH_30_	8.79 ± 0.031	8.28 ± 0.018	8.00 ± 0.092	7.33 ± 0.071	141.19 ± 0.088	130.95 ± 0.055	121.25 ± 0.099	99.8 ± 0.073
NBtH_40_	7.31 ± 0.038	8.32 ± 0.055	8.09 ± 0.050	7.85 ± 0.049	98.87 ± 0.042	129.13 ± 0.091	127.05 ± 0.040	116.71 ± 0.075
	Cinnamic Phenolic Substances—CPS (Caffeic Acid Eq., mg·L^−1^)				Phenolic Flavonoid Substances—PFS (Catechin Eq., mg·L^−1^)			
Aligoté	38.06 ± 0.050				2.13 ± 0.027			
NBt	39.41 ± 0.030	38.73 ± 0.025	40.45 ± 0.072	40.64 ± 0.063	1.99 ± 0.016	1.09 ± 0.024	0.99 ± 0.058	1.12 ± 0.098
NBtH_20_	39.27 ± 0.085	38.19 ± 0.057	39.14 ± 0.070	38.05 ± 0.061	1.94 ± 0.067	1.04 ± 0.063	0.94 ± 0.068	1.02 ± 0.057
NBtH_30_	41.69 ± 0.046	42.02 ± 0.040	40.21 ± 0.062	40.06 ± 0.093	2.00 ± 0.054	1.64 ± 0.044	1.37 ± 0.082	1.00 ± 0.085
NBtH_40_	31.45 ± 0.087	40.21 ± 0.075	39.45 ± 0.092	36.74 ± 0.031	1.20 ± 0.075	1.71 ± 0.035	1.61 ± 0.077	1.29 ± 0.068

## Data Availability

Not applicable.

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
