# Peer review of "Retention of Phthalates in Wine Using Nanomaterials as Chemically Modified Clays with H20, H30, H40 Boltron Dendrimers"

_nanomaterials, 2023, doi:10.3390/nano13162301_

Round 1

Reviewer 1 Report

The objectives are clear and well defined, in this work was to determine the retention of  phthalate-type pollutants and t the protein and  polyphenol levels in the white wine.

The manuscript is original, and represents a good contribution, addition of knowledge to scientific literature of nanomaterial for use in wine industry. Compared with the unmodified na, clay minerals modified with Boltron dendrimers are nanomaterials with diferent structural and morphological properties. But after the analysis of figure 8, doubts remain about the effectiveness of this modification, since the changes are not significant. I believe that the work should better justify the efficacy of the treatments

The strengths of the method described in the manuscript in the experimental results obtained and correct methodology. On another side, the main problem with the work is that does not compare the results with other  purification methods.

 About the change and required information, as follow:

1)The effectiveness of the modifications must be analyzed. It should be indicated to what extent they are significant

2) Table 3 contains an excess of significant figures

3) Is it possible to include electron microscopy photographs?

Author Response

Thank you for your precious time and kind comments on our manuscript entitled “Retention of phthalates in wine using nanomaterials as chemically modified clays with H20, H30, H40 Boltron dendrimers”, which allowed us to improve the manuscript. Also thanks for providing this valuable opportunity to revise our manuscript.

We extremely cherish this opportunity to revise and we have made some revisions compared with the first version of manuscript followed your suggestions and comments and of other reviewers.

Reviewer 2 Report

The authors present results for using clay (sodium bentonite) intercalated with three generations of Boltron dendrimers for removal of pthalic acid esters from wine. Characterization data is presented for the nanomaterials and then results for removal of the PAE from wine. The study is interesting and the paper can be improved by addressing the following:

(1) the chemical structure of Boltron dendrimer should be shown

(2) What are the changes in the basal distance values? Can values be given?

(3) the low wavenumber region of the FTIR spectra are hard to distinguish with respect to the identified peaks. A blow up of the lower wavenumber range would be helpful.

(4)In Figure 7, for this data can an extinction coefficient be noted in the text? Also the y axis range cuts off some of the data.

(5) A reference could be added concerning the phenolic compounds oxidation and the spectra reported in line 347.

(6) Figure 8 - identify the colors in the caption

(7) What is a 'good' value for ITP? Can this be discussed?

There are a small number of spelling errors so please run spell check. Otherwise reads well.

Author Response

(The authors gave the same response as above.)

Reviewer 3 Report

Comments:

The presence of phthalic acid esters in wines presents a major risk to human health, due to their very toxic metabolism. They are known as possible carcinogens. In this paper, aluminosilicate materials were used, to retain various pollutants and unwanted compounds in wine. The pollutants tested are di-butyl and di-ethyl hexyl phthalates. They were tested and detected using the gas chromatography-mass spectrometry (CG-MS) analytical techniques. Nanomaterials were prepared using sodium bentonite, which was chemically modified by impregnation with three types of Boltron dendrimers of second, third, and fourth generations (NBtH20, NBtH30, and NBtH40). The synthesized nanomaterials were characterized using: Brunauer-Emmett-Teller (BET) method, Fourier-transform infrared spectroscopy (FTIR), and X-ray diffraction (XRD) analysis. In this paper, two aspects were followed: the first related to the retention of phthalate-type pollutants (phthalic acid esters - PAEs), and the second related to the protein and polyphenol levels in the white wine of the Aligoté grape variety. The results obtained in this study have a major impact on PAEs in wine, especially after treatment with NBtH30 and NBtH40 (volumes of 250-500 μL/10 mL wine), the retention of the pollutants being up to 85%.

From my side, this is an interesting work. But it still should be carefully revised before it can be considered.

1)      Abstract: “The presence of phthalic acid esters in wines presents a major risk to human health, due to their very toxic metabolism. They are known as possible carcinogens.” Here, “They are known as possible carcinogens.” Can be deleted. Because of the information duplication.

2)      Line 35. “Among the minor compounds in white wine, the most important are: organic acids and phenolic compounds.” Related literature should be provided.

3)      Line 44. Line 49, The paragraph should be text-indented.

4)      Line 33. 3% of what???

5)      Line 152. There should be a blank space “water:ethanol”

6)      Table 1. And table 2. And Table 3. It should better be a three-line table

7)      Figure 6. And Figute 7.This picture is too blurry

8)      There are too many paragraphs in the conclusion. Please simplify the language and highlight the key points.

9)      I do not like this introduction. Please reorganize the introduction to highlight the novelty and contribution of this work.

10)    This background is too small.

11)     The massive production of beer goes hand in hand with the generation of a huge amount of waste during the production process; a greater part of this waste is composed of about 2.6 million tons of Brewers' Spent Grains (BSG) the main waste. Besides, the huge CO2 emissions in the brewers'industry should not be Neglected, because recently, the CO2 emissions are regarded as a bigger problem for human beings and the environment. The authors are suggested to add some information on the CO2 footprint in brewers industry. Hope the literature below can help you.

“[1]Better use of bioenergy: A critical review of co-pelletizing for biofuel manufacturing.Carbon Capture Science & Technology. https://doi.org/10.1016/j.ccst.2021.100003. [2] Minimizing carbon footprint via microalgae as a biological capture.Carbon Capture Science & Technology. https://doi.org/10.1016/j.ccst.2021.100007.[3] Biomethane generation and CO2 recovery through biogas production using brewers' spent Grains. Biocatalysis and Agricultural Biotechnology. https://doi.org/10.1016/j.bcab.2022.102579.” 

12)    The references are too old. Please add more literature in recently 5 years.

13)    Please edit the format of the literature according to the guide for authors.

It should be improved. 

Author Response

Thank you for your precious time and kind comments on our manuscript entitled “Retention of phthalates in wine using nanomaterials as chemically modified clays with H20, H30, H40 Boltron dendrimers”, which allowed us to improve the manuscript. Also thanks for providing this valuable opportunity to revise our manuscript.

We extremely cherish this opportunity to revise and we have made some revisions compared with the first version of manuscript followed your suggestions and comments and of other reviewers.

In response to your concerns and questions, we made the following responses point by point.

Round 2

Reviewer 3 Report

After revision, the quality of this manuscript has been improved. Now it can be considered from my side.